# Blood Transfusions and Adverse Events after Colorectal Surgery: A Propensity-Score-Matched Analysis of a Hen–Egg Issue

**DOI:** 10.3390/diagnostics13050952

**Published:** 2023-03-02

**Authors:** Marco Catarci, Stefano Guadagni, Francesco Masedu, Leonardo Antonio Montemurro, Paolo Ciano, Michele Benedetti, Paolo Delrio, Gianluca Garulli, Felice Pirozzi, Marco Scatizzi

**Affiliations:** 1General Surgery Unit, Sandro Pertini Hospital, ASL Roma 2, 00157 Rome, Italy; 2General Surgery Unit, University of L’Aquila, 67100 L’Aquila, Italy; 3Department of Applied Clinical Sciences and Biotechnology, University of L’Aquila, 67100 L’Aquila, Italy; 4Colorectal Surgical Oncology, Istituto Nazionale per lo Studio e la Cura dei Tumori, “Fondazione Giovanni Pascale IRCCS-Italia”, 80131 Napoli, Italy; 5General Surgery Unit, Infermi Hospital, 47900 Rimini, Italy; 6General Surgery Unit, ASL Napoli 2 Nord, 80078 Pozzuoli (NA), Italy; 7General Surgery Unit, Santa Maria Annunziata & Serristori Hospital, 50012 Firenze, Italy

**Keywords:** blood transfusion, colorectal surgery, transfusion hazards, anastomotic leakage, morbidity, mortality

## Abstract

Blood transfusions are considered a risk factor for adverse outcomes after colorectal surgery. However, it is still unclear if they are the cause (the hen) or the consequence (the egg) of adverse events. A prospective database of 4529 colorectal resections gathered over a 12-month period in 76 Italian surgical units (the iCral3 study), reporting patient-, disease-, and procedure-related variables, together with 60-day adverse events, was retrospectively analyzed identifying a subgroup of 304 cases (6.7%) that received intra- and/or postoperative blood transfusions (IPBTs). The endpoints considered were overall and major morbidity (OM and MM, respectively), anastomotic leakage (AL), and mortality (M) rates. After the exclusion of 336 patients who underwent neo-adjuvant treatments, 4193 (92.6%) cases were analyzed through a 1:1 propensity score matching model including 22 covariates. Two well-balanced groups of 275 patients each were obtained: group A, presence of IPBT, and group B, absence of IPBT. Group A vs. group B showed a significantly higher risk of overall morbidity (154 (56%) vs. 84 (31%) events; OR 3.07; 95%CI 2.13–4.43; *p* = 0.001), major morbidity (59 (21%) vs. 13 (4.7%) events; OR 6.06; 95%CI 3.17–11.6; *p* = 0.001), and anastomotic leakage (31 (11.3%) vs. 8 (2.9%) events; OR 4.72; 95%CI 2.09–10.66; *p* = 0.0002). No significant difference was recorded between the two groups concerning the risk of mortality. The original subpopulation of 304 patients that received IPBT was further analyzed considering three variables: appropriateness of BT according to liberal transfusion thresholds, BT following any hemorrhagic and/or major adverse event, and major adverse event following BT without any previous hemorrhagic adverse event. Inappropriate BT was administered in more than a quarter of cases, without any significant influence on any endpoint. The majority of BT was administered after a hemorrhagic or a major adverse event, with significantly higher rates of MM and AL. Finally, a major adverse event followed BT in a minority (4.3%) of cases, with significantly higher MM, AL, and M rates. In conclusion, although the majority of IPBT was administered with the consequence of hemorrhage and/or major adverse events (the egg), after adjustment accounting for 22 covariates, IPBT still resulted in a definite source of a higher risk of major morbidity and anastomotic leakage rates after colorectal surgery (the hen), calling urgent attention to the implementation of patient blood management programs.

## 1. Introduction

Preoperative anemia is a very common finding, affecting more than 30% of patient candidates for major digestive surgery [1,2]. Consequently, it is the strongest predictor of blood transfusions (five-fold) in the postoperative period [2]. Postoperative anemia affects up to 90% of patients after major surgery [3]. The immediate and most widely used treatment for postoperative anemia is blood transfusion, entailing the risk of several complications, culminating in a higher incidence of morbidity and mortality [4,5,6]. A recent meta-analysis [7] identified blood transfusions (BTs) as a risk factor for poorer early postoperative outcomes, and previous multicenter prospective studies by the Italian The ColoRectal Anastomotic Leakage (iCral) study group [8,9] showed intra- and/or postoperative BT (IPBT) was independently associated with higher morbidity, anastomotic leakage, and mortality rates after colorectal surgery. However, the results of these studies do not allow one to solve the hen–egg issue in which it is still unclear whether blood transfusions are a definite risk factor for poorer outcomes rather than a marker of bad performers: on the one hand, perioperative blood transfusions may induce immunomodulation (transfusion-related immunomodulation, TRIM) because of the infusion of cytokines, lipids, and allogenic leukocytes, leading to immune activation and resulting in transfusion-related acute lung injury (TRALI) or immune suppression, increasing susceptibility to infectious complications; on the other hand, blood transfusions are generally more frequently administered in patients with major comorbidities, more extensive and longer procedures, more advanced cancer stages, and higher intraoperative blood loss. The iCral study group therefore decided to reappraise the results of its last prospective study (iCral3), trying to solve this hen–egg issue.

## 2. Materials and Methods

This is a retrospective analysis of the iCral3 study, designed to assess the influence of adherence to an enhanced recovery pathway (ERP) on patient-reported outcome measures and return to intended oncologic therapy after colorectal surgery. Seventy-six Italian surgical centers voluntarily participated in a prospective enrolment carried out from November 2020 to October 2021, upon explicit inclusion and exclusion criteria [10]. Adherence to twenty-six items of the ERP was measured for each enrolled case upon criteria adapted from the 2018 ERAS Society™ [11] and 2019 national [12] guidelines. For the purposes of this study, the population of 4529 enrolled cases was divided in two groups according to the presence (No. = 304; 6.7%) or absence (No. = 4225; 93.3%) of IPBT. Continuous variables were categorized according to their median value. The Mini Nutritional Assessment—Short Form (MNA-SF [13]) was categorized for values < 12, indicating potential malnutrition. Surgical procedures were categorized as standard (anterior resection, right colectomy, and left colectomy) versus non-standard (splenic flexure resection, transverse colectomy, Hartmann’s reversal, subtotal and total colectomy, and other) resections [9]. Biometric data, patient-, disease-, treatment-, and center-related variables (Table 1) were compared among the two groups using cross tabulation and chi-square or Fisher’s exact test where indicated. All analyses were conducted using StatsDirect™ statistical software (StatsDirect Ltd., Wirral, UK); the significance level was set at *p* < 0.05.

### 2.1. Outcomes

The study endpoints were overall morbidity (OM, any adverse event), major morbidity (MM, any adverse event grade > II according to Clavien-Dindo [14] and the Japanese Clinical Oncology Group (JCOG) extended criteria [15]), anastomotic leakage (AL), defined according to international consensus [16], and mortality (M, any death) rates at 60 days post-surgery.

### 2.2. Propensity-Score-Matched Analysis

Neo-adjuvant therapy is a treatment variable exclusively impacting a subgroup of patients; therefore, to avoid bias in the study design, 336 patients who received a neo-adjuvant treatment were excluded (Figure 1) and a cohort of 4193 cases was divided into two groups according to the presence (Group A; No. = 280; 6.7%) or absence (Group B; No. = 3913; 93.3%) of intra- and/or postoperative blood transfusions (IPBTs). 

The propensity score matching analysis (PSMA) model [19,20] was based on (a) IPBT as the treatment (exposure) variable; (b) group A as the true population of interest; (c) group B as the control population; and (d) the following 22 covariates (confounding variables): sex, age, American Society of Anesthesiologists (ASA) class, body mass index (BMI), diabetes, chronic renal failure, dialysis, chronic liver disease, MNA-SF < 12, surgery for malignancy, urgent admission, preoperative steroids, open approach, standard procedure, associated procedures, operation length, ERP adherence rates, preoperative anemia screening, preoperative BT, hospital type, surgical unit type, and center volume. Adjusted logistic regression was used to estimate the propensity scores in the treatment and control groups. 

Based on the conditioning categorical variables selected, each patient was assigned a propensity score estimated by the standardized mean difference (a standardized mean difference less than 0.1 typically indicates a negligible difference between the means of the groups). No outcome variable was included [21]. As balance is the main goal of PSMA, the analysis was performed using the software “R©” (The R Foundation© for Statistical Computing, Vienna, Austria) with the following specifications: (a) seed 100 for the reproducibility of the analysis; (b) method for distance metric = nearest, distance = logit, caliper = 0.1, replace = false (without sampling replacement), ratio = 1; (c) adjusted logistic regression to estimate the association between the exposure/treatment variable and the outcomes. The following R© libraries/programs were used: “matchit”, “glm”, “publish”, “Tablone”, “Plot”, and “cobalt” [22]. Balance in the matched groups was assessed by calculating the standardized mean difference (SMD) and general variance ratio (a variance ratio close to 1 means that variances are equal in the two groups). For outcome modeling, an adjusted logistic regression, based on IPBT as the treatment variable and on the same 22 covariates selected for the PSMA, was performed, presenting odds ratios (ORs) and their 95% confidence intervals (95%CI). The eventual effect of any unobserved confounder was tested via sensitivity analysis [23], using the R© software library “SensitivityR5” and presenting the Γ values (each 0.1 increment of Γ values representing a 10% odds of differential assignment to treatment due to any unobserved variable).

### 2.3. Subgroup Analysis in the IPBT Population

Considering the population of 304 patients who received one or more IPBT (No. = 304), BT was considered appropriated when administered for Hb levels below liberal [24] transfusion thresholds (≤80 g/L for ASA class I-II, absence of hemodynamic instability, and absence of myocardial ischemia; ≤100 g/L for ASA class III, presence of hemodynamic instability, and/or myocardial ischemia). Furthermore, BT was considered *(the egg)* secondary to bleeding and/or any major adverse event (B/MAE-BT) if it was administered during the operation and/or within 24 h from it, and/or if there was evidence of any previous hemorrhagic (i.e., abdominal bleeding, trocar/wound site bleeding, or anastomotic bleeding) or major adverse event (MAE). Conversely, any MAE was considered *(the hen)* secondary to BT when it occurred after any BT without any previous hemorrhagic adverse event (BT-MAE). Again, these three BT categories were further tested for the endpoints, individually and combined in several scenarios, using cross tabulation and the chi-square or Fisher’s exact test where indicated. All analyses were conducted using StatsDirect™ statistical software (StatsDirect Ltd., Wirral, UK); the significance level was set at *p* < 0.05.

## 3. Results

The outcomes recorded in the whole population are shown in Table 2.

### 3.1. Propensity-Score-Matched Analysis

After propensity score matching, 3643 cases were excluded (5 with IPBT and 3638 without IPBT, and two groups of 275 patients each were generated: group A (IPBT, true population of interest) and group B (no IPBT, control population)). A good balance between the two groups was achieved (Table 3 and Figure 2), with a model variance ratio of 1.005. 

After adjusted logistic regression, group A vs. group B (Table 4) showed a significantly higher risk of OM (154 (56.0%) vs. 84 (30.5%) events; OR 3.07; 95%CI 2.13–4.43; *p* = 0.001), MM (59 (21.4%) vs. 13 (4.7%) events; OR 6.06; 95%CI 3.17–11.6; *p* = 0.001), and AL (31 (11.3%) vs. 8 (2.9%) events; OR 4.72; 95%CI 2.09–10.66; *p* = 0.0002). No difference was recorded between the two groups (8 (2.9%) vs. 5 (1.8%) events; OR 1.57; 95%CI 0.42–5.79; *p* = 0.50) concerning the risk of mortality. 

Compared to local/regional hospitals, metropolitan/academic hospitals showed a significantly lower risk of OM (125/326 (38.3%) vs. 113/224 (50.4%) events; OR 0.61; 95%CI 0.41–0.92; *p* = 0.0166) and mortality (3/326 (0.9%) vs. 10/224 (4.5%) events; OR 0.17; 95%CI 0.03–0.90; *p* = 0.0366). Male vs. female sex was associated with a significantly higher risk of OM (144/309 (46.6%) vs. 94/241 (39.0%) events; OR 1.47; 95%CI 1.0–2.15; *p* = 0.0487) and MM (50/309 (16.2%) vs. 22/241 (9.1%) events; OR 2.26; 95%CI 1.26–4.08; *p* = 0.0066). At the same time, operation length > vs. ≤ 180 min was associated with a significantly higher risk of OM (107/214 (50.0%) vs. 131/336 (39.0%) events; OR 1.60; 95%CI 1.08–2.38; *p* = 0.0183), enrolment > vs. ≤ 44 cases to MM (63/438 (14.4%) vs. 9/112 (8.0%) events; OR 2.36; 95%CI 1.04–5.34; *p* = 0.0397), and presence vs. absence of chronic renal failure to mortality (5/62 (8.1%) vs. 8/488 (1.6%) events; OR 5.11; 95%CI 1.06–24.54; *p* = 0.0416).

### 3.2. Subgroup Analysis in the IPBT Population

Outcome rates according to individual evaluation of the three BT categories (appropriateness; B/MAE-BT; BT-MAE) are shown in Table 5. 

Inappropriate BT was administered in more than a quarter of cases, without any significant influence on any endpoint. On the other hand, the majority of BTs were administered after a hemorrhagic or a major adverse event, with significantly higher rates of MM and AL, but not OM or M. Finally, a BT-MAE was recorded in a minority (4.3%) of cases, showing significantly higher MM, AL, and M rates. Six different scenarios were recorded after matching the three BT categories (Table 6).

All of the scenarios related to BT determined a significant variation in MM, AL, and M rates, with the worst scenario represented by a major adverse event following an appropriate BT.

## 4. Discussion

The comparison of raw data in the subgroups of the whole population (Table 1) fully agrees with the previous findings of the iCral 1 and 2 studies [8,9]; the IPBT subgroup is a reservoir of bad performers (with most of the considered variables showing a significant unfavorable pattern in this subgroup of patients), with significant higher rates of unfavorable outcomes (Table 2). In this setting, it seems that the egg was born before the hen (IPBT may represent the consequence, rather than the cause, of poorer outcomes). Once a nearly perfect balance of the 22 confounding variables was achieved through propensity score matching (Table 3, Figure 2), the paradigm appeared to be totally reversed; the adjusted logistic regression analysis clearly showed (Table 4) that group A (IPBT), compared to group B (no IPBT), is linked to an independent and significant higher risk of OM, MM, and AL (with the lack of statistical significance of the difference concerning the risk of mortality being possibly due to the small number of recorded events). According to these results, it seems that the hen was born before the egg (IPBT may be the cause, rather than the consequence, of poorer outcomes). Assuming that the probabilities of random assignment to the two treatment groups could be different, the sensitivity analysis (Table 4) showed that the relative impact of unknown and/or unmeasured confounding variables should double (Γ = 2.3) for OM and triple (Γ = 3.3) for MM to alter the results and/or their statistical significance. Therefore, the repercussions of this finding on everyday clinical practice are quite relevant: the absolute risk reductions linked to no IPBT recorded in the present study led to small number needed to treat; this could be sufficient to avoid IPBT in 4, 6, and 12 patients to avoid one adverse event, one major adverse event, and one anastomotic leakage, respectively. Another consequence of these findings is that, although the described relationship between blood transfusion and poorer outcomes is not new, a clear understanding of the mechanism by which IPBT may worsen the early outcomes after colorectal surgery is still lacking. Apart from the long-standing and updated concept of TRIM and transient immunosuppression [25,26], a recent retrospective propensity-score-matched study on colorectal cancer surgery patients [27] suggested that the worst early outcomes after surgery for colorectal cancer may be mediated by an exaggerated perioperative systemic inflammatory response in patients receiving perioperative blood transfusions. Moreover, recent experimental evidence [28] suggests a direct link between the gut flora composition (microbiota) and the development of antibody-mediated TRALI in mice. The recent introduction of metabolomics and proteomics to transfusion medicine [29] will possibly clarify how the microbiome and gut microbiota can affect the immune system shaping the antigenicity and contributing to TRIM and the potential transmission of infection by blood donors. As the vast majority of colorectal resections are commonly performed for cancer, representing a particularly vulnerable population and showing significant immunosuppression and altered microbiota [30], further clinical investigation on this issue is warranted. 

Most of the other significant findings of the logistic regression analysis, such as higher risk of adverse outcomes in male vs. female sex, metropolitan/academic vs. local/regional hospitals, operation length > vs. ≤ 180 min, and presence vs. absence of chronic renal failure, were expected, having been already recorded in previous studies [8,9]. On the other hand, the finding of a higher risk of major morbidity recorded in high vs. low volume centers seems to confirm that the surgeon’s volume may be more relevant than center volume [31]. 

Although perioperative BT rates have been declining in the last decade, no change in the risk of mortality after surgery was recorded [32], and there is still a wide variability in perioperative transfusion practices in colorectal surgery [33]. We decided, therefore, to consider liberal (Hb ≤ 80–100 g/L) rather than recommended [24] restrictive (Hb ≤ 70–80 g/L) transfusion thresholds in the analysis of the original subpopulation of patients that received IPBT. Even considering liberal thresholds, inappropriate IPBT was still administered in more than a quarter of cases (Table 5), although this did not determine any significant difference in the outcomes. Anyway, the majority of IPBTs were administered after hemorrhagic and/or major adverse events with a small subgroup of patients (4.3%), in which the BT preceded the major adverse event without any previous hemorrhagic event (Table 5), showing the highest rates of adverse outcomes (Table 6). Applying the long-time-honored 20–80 rule, also known as the “Pareto Principle” [34,35], it could be argued that improving transfusion appropriateness and eliminating this small subgroup of patients may allow for a significant improvement in the outcomes. This is the main aim of the recent call toward the urgent need for patient blood management (PBM) program [36,37] implementation by the World Health Organization [38] and the Italian Surgical Association [39]. Actually, a recent pre- vs. post-PBM implementation study regarding colorectal cancer surgery from Korea [40] showed a significant decrease in the total transfusion rate, Hb threshold before transfusion (Hb trigger), anastomotic leakage rate, and postoperative length of stay. For these reasons, the iCral study group is currently enrolling patients in its fourth observational multicenter prospective study [41], designed to test the effect of adherence to a combined ERP-PBM pathway on blood transfusion rates and outcomes.

The main strength of this study is its methodology: a large database gathered during a prospective multicenter study was analyzed using a PSMA perfectly responding to the EQUATOR (Enhancing the Quality and Transparency of Health Research) network reporting guidelines [18]. Although observational studies cannot be regarded as a replacement for randomized studies, data generated from large observational cohorts have been increasingly used to evaluate important clinical questions where data from randomized trials are limited or do not exist [42], mainly because of lower barriers and cost regarding subject recruitment. PSMA offers an alternative approach for estimating treatment effects with observational data when randomized trials are not feasible or unethical, or when researchers need to assess treatment effects based on real life data, collected through the observation of systems as they operate in normal practice without any intervention implemented via randomized assignment rules, responding to the frequent need to draw conditioned casual inferences from quasi-experimental studies. To account for the conditional probability of treatment selection, thus reducing confounding bias, PSMA presents analytical and interpretation challenges that need to be addressed to maintain the reproducibility of its results, which in recent years has been recognized as a crucial element of high-quality research [43]. The relevant quality of the PSMA used in the present study is based on (1) rigorous patient selection from the parent population, performed adhering to explicit criteria; (2) the inclusion of 22 conditioning variables (covariates), such as hospital type, unit type, and accrual volume, to account for the potential heterogeneity of multicenter, clustered data and adherence to the ERP to account for the potential heterogeneity of medical, anesthesiologic, and surgical perioperative management; (3) a clear, sheer, and restrictive balance algorithm (Figure 1), particularly regarding caliper = 0.1, matching ratio = 1:1, and complete balance assessment; (4) complete description of the software package and of its related analytic details; (5) evaluation of the treatment effect through an adjusted multiple regression model including the same 22 covariates used for matching; and (6) a sensitivity analysis to account for unmeasured confounders.

The other strength of this study is the large number of enrolled patients in a well-defined time-lapse in a large number of centers, representing a very wide sample of surgical units performing colorectal resections in Italy. While the multicenter nature of the parent database may be a definite source of clustering bias, it is undoubtedly representative of real-life data.

However, this study is subject to several limitations, and its results should be interpreted with caution. Several potential confounders were not measured or recorded in the parent study: the number and age of transfused packed red blood cells [44,45], pre- and postoperative Hb levels, iron and Hb status before and after BT [3], the management of preoperative and postoperative anemia through high-dose i.v. iron preparations [46,47], and, as reported above, the composition of blood donors and recipients’ microbiome [48]. Finally, although data quality control was performed and repeated at various levels, we could not rule out potential measurement errors caused by the participating investigators.

## 5. Conclusions

This retrospective PSMA of a large prospective multicenter database confirmed that IPBTs are a definite risk factor for morbidity and anastomotic leakage after colorectal resections even after a well-balanced matching of 22 potential confounders. Although most IPBTs are administered in response to intraoperative blood loss and early postoperative hemorrhagic adverse events, in a minority of cases a major adverse event is triggered by IPBT. In this setting, the avoidance of inappropriate (or unnecessary) BT through the implementation of PBM programs in colorectal surgery may significantly influence the incidence of perioperative adverse outcomes.

## Figures and Tables

**Figure 1 diagnostics-13-00952-f001:**
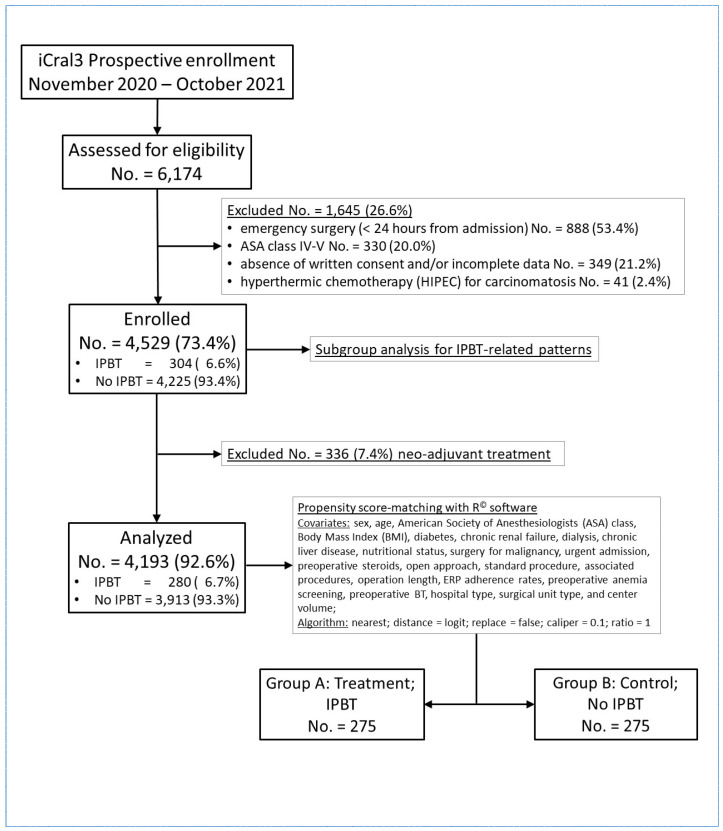
Study flowchart according to the Strengthening the Reporting of Observational Studies in Epidemiology (STROBE) statement guidelines [17] and to the Reporting and Guidelines in Propensity Score Analysis [18]: iCral: Italian ColoRectal Anastomotic Leakage study group; ASA: American Society of Anesthesiologists; IPBT: intra- and/or postoperative blood transfusion(s); and ERP: enhanced recovery pathway.

**Figure 2 diagnostics-13-00952-f002:**
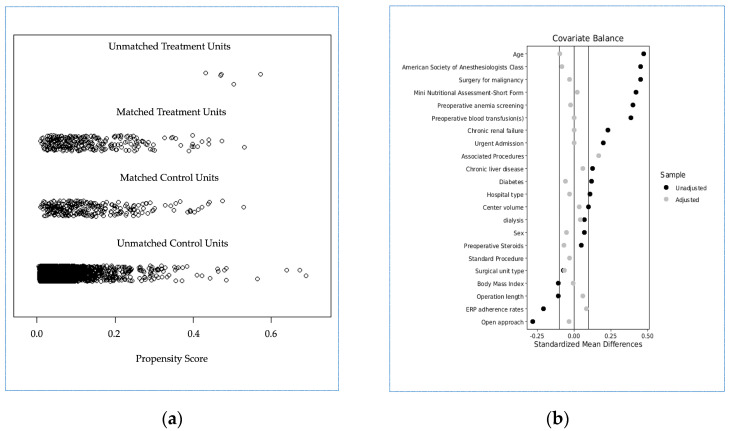
(**a**) Jitter plot distribution of propensity scores in treatment and control groups; (**b**) Love plot of covariates’ standardized mean differences between treatment and control groups before and after matching; the vertical lines represent the interval of ±0.1 and within which balance is considered to be acceptable.

**Table 1 diagnostics-13-00952-t001:** Comparative analysis of study variables in the two groups.

			No ^a^ IPBT (No. = 4225)	^a^ IPBT (No. = 304)	
Variable			No. (%)	No. (%)	*p*
Male sex			2218 (52.5)	171 (56.2)	0.205
Age > 69 years			2014 (47.7)	207 (68.1)	<0.0001
^b^ ASA class III			1373 (32.5)	164 (53.9)	<0.0001
^c^ BMI ≤ 25.0 kg/m^2^			2030 (48.0)	160 (52.6)	<0.0001
Diabetes			573 (13.6)	57 (18.7)	0.014
Chronic renal failure			161 (3.8)	34 (11.2)	<0.0001
Dialysis			7 (0.2)	3 (1.0)	0.025
Chronic liver disease			42 (1.0)	10 (3.3)	0.002
^d^ MNA-SF < 12			1417 (33.5)	164 (53.9)	<0.0001
Disease	Cancer		3021 (71.5)	262 (86.2)	<0.0001
	Diverticular		491 (11.6)	13 (4.3)	
	Endometriosis		165 (3.9)	0 (---)	
	^e^ IBD		143 (3.4)	12 (3.9)	
	Lymphoma		3 (0.1)	1 (0.3)	
	Polyp(s)		223 (5.3)	7 (2.3)	
	Other		179 (4.2)	9 (3.0)	
Elective admission			3970 (94.0)	266 (87.5)	<0.0001
Neo-adjuvant therapy			312 (7.4)	24 (7.9)	0.743
Preoperative steroids			71 (1.7)	8 (2.6)	0.221
Open approach			554 (13.1)	76 (25.0)	<0.0001
Procedure	Right colectomy		1492 (35.5)	172 (56.6)	<0.0001
	Transverse colectomy		87 (2.1)	4 (1.3)	
	Splenic flexure colectomy		123 (2.9)	10 (3.3)	
	Left colectomy		1111 (26.3)	41 (13.5)	
	Anterior resection		964 (22.8)	37 (12.2)	
	^f^ TaTME		44 (1.0)	7 (2.3)	
	Hartmann reversal		102 (2.4)	9 (3.0)	
	(Sub) total colectomy		81 (1.9)	9 (3.0)	
	Other		221 (5.2)	15 (4.9)	
^g^ Standard procedures			3567 (84.4)	250 (82.2)	0.311
Associated procedures			753 (17.8)	78 (25.7)	0.0007
Operation length > 180′			2023 (47.9)	131 (43.1)	0.106
^h^ ERP adherence > median (69.3%)			1902 (45.0)	107 (35.2)	0.001
Preoperative anemia screening			464 (11.0)	88 (29.0)	<0.0001
Preoperative blood transfusion(s)			212 (5.0)	61 (20.1)	<0.0001
Hospital type		Local/Regional	1959 (46.4)	127 (41.8)	0.217
		Metropolitan/Academic	2266 (53.6)	177 (58.2)	
Unit type		General	3564 (84.4)	262 (86.2)	0.006
		Colorectal/Oncologic	661 (15.6)	42 (13.8)	
Center volume		≥44 enrolled cases	3261 (77.2)	245 (80.6)	0.008

^a^ Intra- and/or postoperative blood transfusion(s); ^b^ American Society of Anesthesiologists; ^c^ body mass index; ^d^ Mini Nutritional Assessment—Short Form; ^e^ inflammatory bowel disease; ^f^ transanal total mesorectal excision; ^g^ the sum of right colectomies, left colectomies, and anterior resections; and ^h^ enhanced recovery pathway.

**Table 2 diagnostics-13-00952-t002:** Comparative analysis of outcomes in the two groups.

	No ^a^ IPBT (No. = 4225)	^a^ IPBT (No. = 304)	
Variable	No. (%)	No. (%)	*p*
Overall morbidity	1039 (24.6)	175 (57.6)	<0.0001
Major morbidity	272 (6.4)	69 (22.7)	<0.0001
Anastomotic leakage	167 (3.9)	38 (12.5)	<0.0001
Mortality	52 (1.2)	10 (3.3)	0.008

^a^ Intra- and/or postoperative blood transfusion(s).

**Table 3 diagnostics-13-00952-t003:** Variable distribution in treatment and control groups before and after propensity score matching.

		Before Propensity Score Matching				After Propensity Score Matching			
		^a^ IPBT	No ^a^ IPBT			^a^ IPBT	No ^a^ IPBT		
Variable	Pattern	No. = 280 (6.7%)	No. = 3913 (93.3%)	^b^ p	^c^ SMD	No. = 275 (50.0%)	No. = 275 (50.0%)	^b^ p	^c^ SMD
Sex	Male	154	2017	0.001	1.18	151	158	0.69	0.03
	Female	126	1896	0.001	1.13	124	117	0.66	−0.03
Age (years)	<69	84	2021	0.001	1.26	84	72	0.34	−0.06
	≥69	196	1892	0.001	1.06	191	203	0.49	0.05
^d^ ASA class	I-II	125	2630	0.001	1.65	125	114	0.46	−0.05
	III	155	1283	0.001	0.76	150	161	0.50	0.04
^e^ BMI (kg/m^2^)	≤25	149	1873	0.001	1.10	145	144	1.00	−0.004
	>25	131	2040	0.001	1.22	130	131	1.00	0.004
Diabetes	Yes	51	532	0.001	0.46	50	56	0.61	0.04
	No	229	3381	0.001	2.33	225	219	0.76	−0.02
^f^ CRF	Yes	31	152	0.001	0.20	31	31	1.00	0.00
	No	249	3761	0.001	3.08	244	244	1.00	0.00
Dialysis	Yes	2	5	0.45	0.02	2	1	1.00	−0.03
	No	278	3908	0.001	3.46	273	274	1.00	0.004
^g^ CLD	Yes	9	38	0.001	0.09	7	4	0.54	−0.05
	No	271	3875	0.001	3.36	268	271	0.90	0.01
^h^ MNA-SF	≤12	128	2609	0.001	1.63	128	131	0.89	0.01
	>12	152	1304	0.001	0.78	147	144	0.89	−0.01
Cancer	Yes	239	2713	0.001	1.57	234	237	0.90	0.01
	No	41	1200	0.001	0.85	41	38	0.82	−0.02
Admission	Elective	243	3661	0.001	2.83	239	239	1.00	0.00
	Urgent	37	252	0.001	0.28	36	36	1.00	0.00
Preoperative steroids	Yes	7	67	0.001	0.15	7	10	0.62	0.04
	No	273	3846	0.001	3.26	268	265	0.90	−0.01
^j^ MI surgery	Yes	209	3396	0.001	2.40	207	211	0.85	0.01
	No	71	517	0.001	0.43	68	64	0.78	−0.02
Standard procedure	Yes	233	3298	0.001	2.20	229	232	0.90	0.01
	No	47	615	0.001	0.52	46	43	0.83	−0.02
Associated procedures	Yes	71	705	0.001	0.54	69	49	0.06	−0.12
	No	209	3208	0.001	2.12	206	226	0.24	0.07
Operation length (min.)	≤180	167	2128	0.001	1.23	164	172	0.65	0.03
	>180	113	1785	0.001	1.08	111	103	0.59	−0.04
^k^ ERP adherence (%)	≤69.3	181	2142	0.001	1.23	177	188	0.52	0.04
	>69.3	99	1771	0.001	1.09	98	87	0.42	−0.05
^l^ Preop AS	Yes	83	445	0.001	0.36	79	82	0.86	0.02
	No	197	3468	0.001	2.55	196	193	0.90	−0.01
^m^ Preop BT	Yes	59	206	0.001	0.20	55	55	1.00	0.00
	No	221	3707	0.001	3.01	220	220	1.00	0.00
Hospital type	^n^ LR	117	1846	0.001	1.11	114	110	0.82	−0.02
	^o^ MA	163	2067	0.001	1.20	161	165	0.84	0.02
Unit type	^p^ GS	245	3331	0.001	2.23	240	234	0.76	−0.02
	^q^ CO	35	582	0.001	0.52	35	41	0.55	0.04
Enrolment (no. of cases)	≤44	54	909	0.001	0.68	54	58	0.76	0.02
	>44	226	3004	0.001	1.86	221	217	0.85	−0.01
^r^ OM	Yes	158	944	0.001	0.58	154	84	0.01	−0.31
	No	122	2969	0.001	1.98	121	191	0.01	0.29
^s^ MM	Yes	60	236	0.001	0.23	59	13	0.001	−0.34
	No	220	3677	0.001	2.94	216	262	0.006	0.17
^t^ AL	Yes	31	135	0.001	0.18	31	8	0.001	−0.23
	No	249	3778	0.001	3.13	244	267	0.18	0.08
Mortality	Yes	9	48	0.001	0.11	8	5	0.58	−0.05
	No	271	3865	0.001	3.33	267	270	0.90	0.01

^a^ Intra- and/or postoperative blood transfusion(s); ^b^ Student’s test for proportions; ^c^ standardized mean difference ^d^ American Society of Anesthesiologists; ^e^ body mass index; ^f^ chronic renal failure; ^g^ chronic liver disease; ^h^ Mini Nutritional Assessment—Short Form; ^j^ Mininvasive; ^k^ enhanced recovery pathway; ^l^ preoperative anemia screening; ^m^ preoperative blood transfusion(s); ^n^ local/regional; ^o^ metropolitan/academic; ^p^ general surgery; ^q^ colorectal/oncologic surgery; ^r^ overall morbidity; ^s^ major morbidity; and ^t^ AL: anastomotic leakage.

**Table 4 diagnostics-13-00952-t004:** Adjusted multiple regression analysis for endpoints.

		Overall Morbidity		Major Morbidity		Overall ^a^ AL		Mortality	
Variable	Pattern	^b^ OR (95%CI)	*p*	^b^ OR (95%CI)	*p*	^b^ OR (95%CI)	*p*	^b^ OR (95%CI)	*p*
^c^ IPBT	Yes	3.07 (2.13–4.43)	0.001	6.06 (3.17–11.6)	0.001	4.72 (2.09–10.66)	0.0002	1.57 (0.42–5.79)	0.50
	No	Reference		Reference		Reference		Reference	
Sex	Male	1.47 (1.00–2.15)	0.05	2.26 (1.26–4.08)	0.007	1.24 (0.60–2.56)	0.56	3.56 (0.69–18.42)	0.13
	Female	Reference		Reference		Reference		Reference	
Age (years)	<69	Reference		Reference		Reference		Reference	
	≥69	0.96 (0.62–1.51)	0.87	0.76 (0.40–1.46)	0.41	0.87 (0.37–2.04)	0.74	4.81 (0.42–55.12)	0.21
^d^ ASA class	I-II	Reference		Reference		Reference		Reference	
	III	1.02 (0.67–1.54)	0.94	1.10 (0.59–2.06)	0.76	1.88 (0.82–4.31)	0.13	1.63 (0.30–8.72)	0.57
Body mass index (kg/m^2^)	≤25	Reference		Reference		Reference		Reference	
	>25	0.92 (0.62–1.36)	0.68	0.97 (0.54–1.72)	0.91	0.97 (0.47–2.02)	0.94	0.43 (0.10–1.84)	0.26
Diabetes	Yes	0.84 (0.52–1.35)	0.47	0.46 (0.20–1.04)	0.06	0.68 (0.25–1.83)	0.45	0.19 (0.02–2.10)	0.18
	No	Reference		Reference		Reference		Reference	
Chronic renal failure	Yes	0.81 (0.43–1.51)	0.51	1.49 (0.63–3.53)	0.36	1.41 (0.50–3.98)	0.52	5.11 (1.06–24.54)	0.04
	No	Reference		Reference		Reference		Reference	
Dialysis	Yes	1.97 (0.16–24.8)	0.60	1.52 (0.10–23.6)	0.77	Not Estimable		7.15 (0.15–344.47)	0.32
	No	Reference		Reference				Reference	
Chronic liver disease	Yes	0.61 (0.16–2.36)	0.47	0.58 (0.06–5.29)	0.63	Not Estimable		Not Estimable	
	No	Reference		Reference					
^e^ MNA-SF	≤12	Reference		Reference		Reference		Reference	
	>12	1.46 (0.99–2.14)	0.053	0.80 (0.46–1.41)	0.45	1.57 (0.75–3.27)	0.23	2.21 (0.47–10.40)	0.32
Surgery for malignancy	Yes	1.22 (0.69–2.17)	0.50	1.42 (0.60–3.37)	0.42	0.77 (0.28–2.13)	0.62	0.51 (0.09–3.01)	0.46
	No	Reference		Reference		Reference		Reference	
Admission	Elective	Reference		Reference		Reference		Reference	
	Urgent	0.88 (0.49–1.56)	0.66	0.81 (0.33–2.00)	0.65	0.77 (0.24–2.47)	0.66	0.70 (0.11–4.69)	0.72
Preoperative steroids	Yes	1.13 (0.39–3.33)	0.82	0.31 (0.03–2.82)	0.30	0.68 (0.07–6.11)	0.73	3.75 (0.26–53.80)	0.33
	No	Reference		Reference		Reference		Reference	
^f^ MI surgery	Yes	0.86 (0.54–1.37)	0.52	0.73 (0.36–1.50)	0.39	1.10 (0.44–2.73)	0.84	0.50 (0.09–2.77)	0.43
	No	Reference		Reference		Reference		Reference	
Standard procedures	Yes	1.25 (0.75–2.10)	0.39	0.51 (0.26–1.02)	0.06	0.78 (0.31–1.96)	0.60	1.38 (0.23–8.23)	0.73
	No	Reference		Reference		Reference		Reference	
Associated procedures	Yes	0.87 (0.55–1.38)	0.56	0.56 (0.27–1.15)	0.11	0.44 (0.16–1.17)	0.10	0.68 (0.14–3.32)	0.63
	No	Reference		Reference		Reference		Reference	
Operation length (min.)	≤180	Reference		Reference		Reference		Reference	
	>180	1.60 (1.08–2.38)	0.02	1.07 (0.60–1.90)	0.83	1.92 (0.92–3.98)	0.08	2.25 (0.56–9.08)	0.26
^g^ ERP adherence (%)	≤69.3	Reference		Reference		Reference		Reference	
	>69.3	1.30 (0.84–2.01)	0.24	1.10 (0.58–2.08)	0.76	0.59 (0.25–1.41)	0.24	0.60 (0.13–2.87)	0.52
^h^ Preop AS	Yes	1.36 (0.78–2.39)	0.28	1.40 (0.63–3.09)	0.41	1.53 (0.54–4.34)	0.42	0.74 (0.09–6.05)	0.78
	No	Reference		Reference		Reference		Reference	
^i^ Preop BT	Yes	0.92 (0.49–1.69)	0.78	0.77 (0.32–1.84)	0.55	0.54 (0.16–1.80)	0.31	3.28 (0.47–22.68)	0.23
	No	Reference		Reference		Reference		Reference	
Hospital type	^l^ LR	Reference		Reference		Reference		Reference	
	^m^ MA	0.61 (0.41–0.92)	0.02	0.81 (0.45–1.48)	0.50	1.06 (0.48–2.33)	0.88	0.17 (0.03–0.90)	0.04
Unit type	^n^ GS	Reference		Reference		Reference		Reference	
	^o^ CO	1.04 (0.58–1.85)	0.90	1.00 (0.42–2.37)	0.99	0.86 (0.28–2.65)	0.79	0.90 (0.07–12.12)	0.94
Enrolment (no. of cases)	≤44	Reference		Reference		Reference		Reference	
	>44	0.84 (0.52–1.37)	0.49	2.36 (1.04–5.34)	0.04	1.90 (0.69–5.20)	0.21	1.99 (0.30–13.30)	0.48
**Sensitivity analysis**		**Γ**		**Γ**		**Γ**		**Γ**	
		2.3	0.051	3.3	0.05	2.3	0.06	1	0.29

^a^ AL: anastomotic leakage; ^b^ OR (95%CI): odds ratio and 95% confidence intervals; ^c^ intra- and/or postoperative blood transfusions; ^d^ ASA: American Society of Anesthesiologists; ^e^ Mini Nutritional Assessment—Short Form; ^f^ Mininvasive; ^g^ ERP: enhanced recovery pathway; ^h^ preoperative anemia screening; ^i^ preoperative blood transfusion(s); ^l^ local/regional; ^m^ metropolitan/academic; ^n^ general surgery; and ^o^ colorectal/oncologic surgery.

**Table 5 diagnostics-13-00952-t005:** Outcome rates according to individual BT categories.

	Endpoint	Overall Morbidity		Major Morbidity		Anastomotic Leakage		Mortality	
^a^ BT Category	Pattern No. (%)	No. (%)	*p*	No. (%)	*p*	No. (%)	*p*	No. (%)	*p*
Appropriateness	Yes 225 (74.0)	131 (58.2)	0.845	51 (22.7)	0.983	28 (12.4)	0.391	10 (4.4)	0.07
	No 79 (26.0)	45 (57.0)		18 (22.8)		7 (8.9)		0 (- -.-)	
^b^ B/MAE-BT	Yes 155 (51.0)	89 (57.4)	0.864	52 (33.5)	<0.001	24 (15.5)	0.027	6 (3.9)	0.751
	No 149 (49.0)	87 (58.4)		17 (11.4)		11 (7.4)		4 (2.7)	
^c^ BT-MAE	Yes 13 (4.3)	10 (76.9)	0.250	12 (92.3)	<0.0001	5 (38.5)	0.001	2 (15.4)	0.063
	No 291 (95.7)	166 (57.0)		57 (19.6)		26 (10.3)		8 (2.7)	

^a^ Blood transfusion(s); ^b^ bleeding- and/or major-adverse-event-related blood transfusion(s); and ^c^ blood-transfusion(s)-related major adverse event.

**Table 6 diagnostics-13-00952-t006:** Matching scenarios of BT categories.

Scenario		Overall Morbidity		Major Morbidity		Anastomotic Leakage		Mortality	
	No. (%)	No. (%)	^α^ *p*	No. (%)	^α^ *p*	No. (%)	^α^ *p*	No. (%)	^α^ *p*
Appropriate ^a^ BT No ^b^ B/MAE-BT No ^c^ BT-MAE	99 (32.6)	56 (56.6)	0.746	4 (4.0)	<0.0001	3 (3.0)	0.0003	2 (2.0)	0.0026
Inappropriate ^a^ BT No ^b^ B/MAE-BT No ^c^ BT-MAE	37 (12.2)	21 (56.7)		2 (5.4)		3 (8.1)		0 (-.-)	
Appropriate ^a^ BT ^b^ B/MAE-BT No ^c^ BT-MAE	120 (39.5)	70 (58.3)		40 (33.3)		22 (18.3)		6 (5.0)	
Inappropriate ^a^ BT ^b^ B/MAE-BT No ^c^ BT-MAE	35 (11.5)	19 (66.7)		12 (34.3)		2 (5.7)		0 (-.-)	
Inappropriate ^a^ BT No ^b^ B/MAE-BT ^c^ BT-MAE	6 (2.0)	4 (66.7)		4 (66.7)		2 (16.7)		0 (-.-)	
Appropriate ^a^ BT No ^b^ B/MAE-BT ^c^ BT-MAE	7 (2.3)	6 (85.7)		7 (100.0)		3 (42.8)		2 (28.6)	

^α^ Two by six chi-square test with five degrees of freedom; ^a^ blood transfusion(s); ^b^ bleeding- and/or major-adverse-event-related blood transfusion(s); and ^c^ blood-transfusion(s)-related major adverse event.

## Data Availability

Individual participant-level anonymized datasets are available upon reasonable request by contacting the study coordinator.

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
