# Peer review of "Blood Transfusions and Adverse Events after Colorectal Surgery: A Propensity-Score-Matched Analysis of a Hen–Egg Issue"

_diagnostics, 2023, doi:10.3390/diagnostics13050952_

Round 1

Reviewer 1 Report

Thank you for this interesting manuscript looking into the effects of blood transfusion in patients undergoing colorectal surgery. It has always been hard to tease out the cause and effect relationship between blood transfusions and poor outcomes. This manuscript does a good job in defining these differences. I would be interested to look into the group of patients receiving neoadjuvant therapy who were excluded as many of our rectal cancers would receive this as standard of care. I am curious as to how many patients with anterior resections had lower stage rectal cancers not requiring neoadjuvant treatment were included since these would tend to be less technically challenging than a more bulky rectal cancer that would need neoadjuvant treatment as standard of care. 

Author Response

Reviewer #1

Thank you for this interesting manuscript looking into the effects of blood transfusion in patients undergoing colorectal surgery. It has always been hard to tease out the cause and effect relationship between blood transfusions and poor outcomes. This manuscript does a good job in defining these differences. I would be interested to look into the group of patients receiving neoadjuvant therapy who were excluded as many of our rectal cancers would receive this as standard of care. I am curious as to how many patients with anterior resections had lower stage rectal cancers not requiring neoadjuvant treatment were included since these would tend to be less technically challenging than a more bulky rectal cancer that would need neoadjuvant treatment as standard of care. 

Thank you for your comments. The vast majority (216 out of 336; 64.3%) of cases excluded from the PSM analysis because submitted to neoadjuvant treatment was composed by locally advanced rectal cancer (i.e. cT>2 and/or cN+) located within 10 cm from the external anal verge. On the other hand, the population of 550 cases included in the PSM analysis included 60 cases (10.9%) of anterior resection for rectal cancer, of which 20 (33%) were located within 10 cm from the external anal verge with earlier stages at preoperative staging. There is still a lot of controversy regarding the outcomes following surgery for rectal cancer after neoadjuvant therapy [1]. On one hand, the reviewer is perfectly right, as the potential for dense fibrosis and advanced stage may make surgery and related intraoperative blood loss definitely more challenging in this subset of patients. On the other hand, it has been clearly demonstrated that neoadjuvant therapy can induce tumour regression and partial or complete pathological response, making the surgical approach somewhat easier [2]. Anyway, the methodology of PSM analysis [3] requires exclusion of any treatment confounder (i.e. administered to a minority of cases) that may add significant bias. Accordingly, all cases submitted to neoadjuvant treatment were excluded.

1] Hu MH, Huang RK, Zhao RS, Yang KL, Wang H. Does neoadjuvant therapy increase the incidence of anastomotic leakage after anterior resection for mid and low rectal cancer? A systematic review and meta-analysis. Colorectal Dis. 2017 Jan;19(1):16-26.

2] van Gijn W, Marijnen CA, Nagtegaal ID, et al.Preoperative radiotherapy combined with total mesorectal excision for resectable rectal cancer: 12-year follow-up of the multicentre, randomised controlled TME trial. Lancet Oncol. 2011;12:575–82.

3] Yao XI, Wang X, Speicher PJ, et al. Reporting and Guidelines in Propensity Score Analysis: A Systematic Review of Cancer and Cancer Surgical Studies. J Natl Cancer Inst. 2017;109(8):djw323.

Reviewer 2 Report

Congrats to the authors.

Author Response

Reviewer #2

Although the association of transfusions with the occurrence of complications after various types of surgery is a much-debated topic, the way the authors approach it [through a 1:1 propensity score-matching model, including 22 covariates] makes it extremely interesting.

Major Comments

Since there has been a lot of discussion in recent years about the relationship between gut microbiome and blood transfusions, I would like the authors to include, either in the discussion section or in the limitations of the study a comment to the role of blood transfusions in the alteration of the microbiome and the post-op complications that could be associated with it. I understand that the study protocol has a different orientation, and data from the patients’ microbiome is difficult to have been collected, although it would be of great interest to have such a study

This is a very good point. Unfortunately, as underlined by the reviewer, the influence of blood transfusions on gut microbiome/microbiota was not investigated nor recorded per protocol, and related data is therefore simply not available. Anyway, we keep this important suggestion for future studies on this topic. Pertinent comment on this issue was added in the discussion about mechanisms by which blood transfusions may be related to a higher risk of worse outcomes as well as in the limitations of the study.

Minor Comments

Authors should, in my opinion, avoid such an extensive use of parentheses, which contain up to 3 printed lines of text. see lines 59 to 62, 62 to 64, 244 to 246, 265 to 268

All the related sentences were rephrased accordingly.

Reviewer 3 Report

Although the association of transfusions with the occurrence of complications after various types of surgery is a much-debated topic, the way the authors approach it [through a 1:1 propensity score-matching model, including 22 covariates] makes it extremely interesting.

Major Comments

Since there has been a lot of discussion in recent years about the relationship between gut microbiome and blood transfusions, I would like the authors to include, either in the discussion section or in the limitations of the study a comment to the role of blood transfusions in the alteration of the microbiome and the post-op complications that could be associated with it. I understand that the study protocol has a different orientation, and data from the patients’ microbiome is difficult to have been collected, although it would be of great interest to have such a study

Minor Comments

Authors should, in my opinion, avoid such an extensive use of parentheses, which contain up to 3 printed lines of text. see lines 59 to 62, 62 to 64, 244 to 246, 265 to 268

Author Response

Reviewer #3

In this study, Marco Catarci, et al. studied the risk of intra- and/or post operative blood transfusions on overall and major morbidity, anastomotic leakage, and mortality rates after colorectal surgery. Authors conclude that IPBT is a definite risk factor for morbidity and anastomotic leakage after colorectal resections. The study question is valid and very relevant in clinical practice. The manuscript is well written and study findings support authors conclusions.

The manuscript can be improved by addressing following minor concerns:

-       Spell out the terms MM and AL in the abstract.

Sorry for this unacceptable style mismatch. The shorthands were quoted in the abstract.

-       Introduction section of the manuscript does not discuss about the context and significance of this study in detail. Consider adding more information.

We revised the introduction accordingly, hoping it to give now a clear outoline of the context and significance of our study.

-       The study does not address the mechanism of relationship between blood transfusion and poor postoperative outcomes. Authors may want to add some speculations in discussion.

Good point. According to point #1 of reviewer #2, a paragraph describing the the mechanism of relationship between blood transfusion and poor postoperative outcomes was added in the discussion.

Reviewer 4 Report

In this study, Marco Catarci, et al. studied the risk of intra- and/or post operative blood transfusions on overall and major morbidity, anastomotic leakage, and mortality rates after colorectal surgery. Authors conclude that IPBT is a definite risk factor for morbidity and anastomotic leakage after colorectal resections. The study question is valid and very relevant in clinical practice. The manuscript is well written and study findings support authors conclusions.

The manuscript can be improved by addressing following minor concerns:

-       Spell out the terms MM and AL in the abstract.

-       Introduction section of the manuscript does not discuss about the context and significance of this study in detail. Consider adding more information.

-       The study does not address the mechanism of relationship between blood transfusion and poor postoperative outcomes. Authors may want to add some speculations in discussion.

Author Response

(The authors gave the same response as above.)
